# Theragnostic Aspects and Radioimmunotherapy in Pediatric Tumors

**DOI:** 10.3390/ijms21113849

**Published:** 2020-05-28

**Authors:** Andrea Cimini, Maria Ricci, Agostino Chiaravalloti, Luca Filippi, Orazio Schillaci

**Affiliations:** 1Department of Biomedicine and Prevention, University Tor Vergata, 00133 Rome, Italy; maria.ricci28@gmail.com (M.R.); agostino.chiaravalloti@gmail.com (A.C.); orazio.schillaci@uniroma2.it (O.S.); 2Nuclear Medicine Section, IRCCS Neuromed, 86077 Pozzilli, Italy; 3Nuclear Medicine Section, “Santa Maria Goretti” Hospital, 04100 Latina, Italy; lucfil@hotmail.com

**Keywords:** theragnostics, radioimmunotherapy, pediatric tumors, neuroblastoma, pediatric brain tumors, nuclear medicine, positron emission tomography, single-photon emission computed tomography, personalized therapy, radiopharmaceuticals

## Abstract

The use of theragnostic radiopharmaceuticals in nuclear medicine has grown rapidly over the years to combine the diagnosis and therapy of tumors. In this review, we performed web-based and desktop literature research to investigate and explain the potential role of theragnostic imaging in pediatric oncology. We focused primarily on patients with aggressive malignancies such as neuroblastoma and brain tumors, to select patients with the highest chance of benefit from personalized therapy. Moreover, the most critical and groundbreaking applications of radioimmunotherapy in children’s oncology were examined in this peculiar context. Preliminary results showed the potential feasibility of theragnostic imaging and radioimmunotherapy in pediatric oncology. They revealed advantages in the management of the disease, thereby allowing an intra-personal approach and adding new weapons to conventional therapies.

## 1. Introduction

Precision medicine in cancer is a rapidly growing field that developed out of an improved understanding of the genetic basis of disease and the characterization of tumor phenotypes. Through the identification of individual tumor molecular and genetic signatures, the expectation is that treatment can be tailored to patient subpopulations or individual patients [1]. Precision medicine is also rapidly growing in the pediatric oncology field, where a personalized approach may lead to an improvement in management. Actual goals in pediatric oncology include the achievement of therapeutic scope and to avoid damage to healthy tissue and critical organs. Radiotherapy with tumor-targeted radionuclides may overcome some of these challenges. There exist several cancer-selective agents used for pediatric oncology, which depend on the histopathologic origin of tumors [2].

Moreover, the importance of personalized medicine has been growing. Several nuclear medicine techniques allow the evaluation of specific molecular targets both for diagnostics and for therapy purposes. The diagnostic approach evaluates potential targets that can predict if a patient will benefit from a particular treatment. Increasing literature on the genetic and molecular aspects of tumor subtypes describes the intra-personal variability and the importance of the development of potentially effective targeted therapies [3]. Therefore, rapid growth in the radiopharmaceuticals and diagnostic fields has led to the continually increasing use of theragnostic agents [4].

Positron emission tomography (PET) traces can be used to predict the behavior of radioactive or non-radioactive targeted drugs (i.e., radiopharmaceuticals or pharmaceuticals). In the case of pharmaceuticals, the role of nuclear medicine tracers is to target the pharmaceutical compound biodistribution [3].

In the case of radiopharmaceuticals, to assess biodistribution and allow dosimetric analysis, targeted drugs are labeled with a radionuclide for use in a diagnostic procedure. Several compounds may be marked with radionuclides, with chemical, immunological, or molecular targets. Moreover, developments in the innovative field of radioimmunotherapy are noteworthy. Radiolabeled antibodies, combining the effects of immunotherapy and radiotherapy, provide unique strategies in cancer treatment [5]. The cross-fire effect, led by alpha or beta-emitters radioisotopes, enhances the therapeutic effect of immunotherapy killing nearby tumor cells that are not bound by antibodies [6].

In most cases, a pre-therapeutic assessment is performed. The same drug can be used both in diagnostic/pre-therapeutic assessment and in theragnostic, based on the physical properties of the labeled radionuclide. Generally, the same targeting drug is marked with a gamma-emitting radionuclide for the pre-therapeutic assessment and with alpha or beta particle-emitting radionuclide for theragnostic purposes. Moreover, the same radionuclide can be used for both scopes with a reduced dose for the pre-therapeutic assessment. The pre-therapeutic evaluation leads to a personalized approach that includes the evaluation of drug biodistribution before the treatment and, eventually, provisional dosimetry.

The pre-therapeutic assessment influences the management of the patients. The dosimetric assessment on tumor and critical organs appears particularly useful in the pediatric population to avoid ineffective treatments and side-effects, primarily on essential organs. If recommended by the diagnostic imaging pattern, the same targeting drug, labeled with an alpha (α-) or beta (β-) particle-emitting radionuclide, is administered to induce localized DNA double-strand breaks and cell death. [3]. Several isotopes are used in imaging and therapy, according to their physical and chemical properties. Generally, for imaging purposes, γ emission is preferred (iodine 123 (^123^I), indium 111 (^111^In)).

Regarding iodine-binding drugs, the iodine 131 (^131^I) is the selected isotope for therapy purposes due to the cytocidal effect on the cells of the β rays emitted [7]. ^123^I is used more for imaging owing to its short half-life and the γ emission ideal for gamma camera imaging, with a lack of β emission.

However, given the limited availability of ^123^I, ^131^I can be used for imaging (with a reduced dose) and therapy in adults [8]. In children, because of the cytocidal effects of ^131^I emission, other iodine isotopes are generally indicated for imaging aims. ^124^I has the potential to improve the current clinical practice in the diagnosis and dosimetry of malignant diseases such as neuroblastoma, paraganglioma, pheochromocytoma, and carcinoids with an elaborate radioactive decay scheme, which includes several high-energy gamma rays [9,10].

Gallium 68(^68^Ga)-labeled pharmaceuticals have the potential to cover several of today’s clinical options with gamma camera imaging isotopes with a concordant higher resolution of PET in comparison to gamma camera imaging [11,12].

The ^68^Ga-based diagnostic is utilized to determine if the biological target is present and, if so, a therapeutic isotope (e.g., Lutetium 177 (^177^Lu), actinium (^225^Ac)) can be complexed with the same scaffold to generate a corresponding radiotherapeutic [13]. Particularly, peptide receptor radionuclide therapy (PRRT) with ^177^Lu and yttrium 90 (^90^Y) of patients with somatostatin receptor-expressing neuroendocrine neoplasms has shown promising results [14]

There remain several unresolved clinical questions as to the optimization of diagnostic techniques and dosimetry protocols. This review aims to focus on the novel tracers and innovative developments in the theragnostic field in children. Moreover, we aim to explore the role of pre-therapy and dosimetry in theragnostic to discuss the impact of personalized medicine in pediatric oncology. An overview of research articles concerning pediatric theragnostic and radioimmunotherapy is reported in Table 1. Moreover, an overview of the main radiopharmaceuticals is summarized in Table 2.

## 2. Theragnostic Tracers in Neuroectodermal Tumors

### 2.1. Metaiodobenzylguanidine (MIBG)

MIBG is a noradrenaline (norepinephrine) analog and so-called “false” neurotransmitter. This radiopharmaceutical, labeled with ^131^I, could be used as a radiotherapeutic metabolic agent in neuroectodermal tumors (NETs). The tumors are derived from the primitive neural crest, which develops to form the sympathetic nervous system [8].

Iodine–labeled MIBG is a functional agent ideal for use in imaging of neuroendocrine tumors. Therefore, MIBG diagnostic and theragnostic procedures are used for localization, staging, and in follow-up and eventual therapy of NETs like pheochromocytoma, neuroblastoma, ganglioneuroblastoma, paragangliomas, medullary carcinoma of the thyroid, carcinoid, Merkel cell tumors, and MEN (multiple endocrine neoplasia) syndrome, and as a prelude to ^131^I MIBG therapy [7]. Being a systemic agent labeled with the β emitter ^131^I, the therapeutic efficacy of ^131^I MIBG to multiple sites in one sitting is higher than that of the external beam radiation therapy. [7].

Pediatric MIBG theragnostic approach is focused mainly on neuroblastoma, which is the third most common childhood cancer and the most common extracranial solid tumor of childhood, with an aggressive nature and high likelihood of metastatic disease. As a norepinephrine analog, MIBG uptake is allowed by norepinephrine transporters [35]. This outcome is shown in the majority of neuroblastomas (up to 90%) [36].

The general data requirement for ^131^I MIBG treatment includes a proven, inoperable neuroendocrine tumor. Eligible patients will have MIBG-positive tumors, documented by tracer scintigraphy using ^123^I-MIBG as a diagnostic tracer (in adults also ^131^I-MIBG). Tumor-absorbed doses vary widely, and thus, provide useful information regarding treatment efficacy [8,37].

To perform therapy with MIBG, the pre-therapeutic MIBG labeled with ^123^I (or ^131^I in adults) scanning should demonstrate a high uptake of the radiopharmaceutical in primary and/or secondary sites. [7]. Semiquantitative scoring of skeletal segments such as the modified Curie score or the International Society of Paediatric Oncology Europe Neuroblastoma (SIOPEN) score is used to assess MIBG avidity [35]. Figure 1 shows a pre-therapeutic staging in a child with neuroblastoma, performed with ^123^I MIBG scintigraphy/single-photon emission computed tomography (SPECT).

Possible collateral effects of MIBG theragnostic therapy are mainly due to the variable degree of bone marrow toxicity and to rare but possible long-term effects of ^131^I therapy (i.e., hypothyroidism, persistent hematological effects, and extremely rare evidence for induction of leukemia or secondary solid tumors). In this regard, a correlation between ^131^I-MIBG treatment dose per body weight and hematological toxicity has been reported in a dose-escalation study conducted by Matthay et al. involving 30 patients with neuroblastoma [38]. The pre-therapeutic dosimetry scanning can be used clinically to determine the predicted level of uptake and retention from a therapeutic administration, thereby predicting the whole-body radiation dose and the bone marrow radiation dose and, subsequently, the individual degree of bone marrow toxicity [8]. It is worth mentioning the interesting results achieved in high-risk neuroblastoma by combining high dose chemotherapy and radionuclide targeted therapy. As early as 2001, Mastrangelo et al. evaluated the efficacy and safety of the combined therapy with ^131^I-MIBG and chemotherapy in a 10-day course in 16 children with resistant and relapsed neuroblastoma [39]. The majority of patients (12 of 16) showed a partial response and the only toxicity demonstrated was hematological (neutropenia and thrombocytopenia). The same authors showed the efficacy and the feasibility of ^131^I-MIBG chemotherapy combination treatment in 13 pediatric patients with newly diagnosed neuroblastoma as well [40]. Forty days after the beginning of combined therapy, the responses were excellent in the majority of cases (2 children showed a complete response, six children had very good partial responses) and hematological toxicity was acceptable, even with extensively infiltrated bone marrow.

Moreover, Suh et al. treated neuroblastoma patients with tandem high dose chemotherapy/autologous stem cell transplantation, plus ^131^I-MIBG therapy (dose ranging from 395.9 Megabecquerels(MBq)/kg to 636.4 MBq/kg). The authors reported an overall survival of 83% after five years. Of note, no treatment-related mortality was registered [15]. High dose ^131^I-MIBG targeted therapy has been applied recently in Japan to treat children with relapsed neuroblastoma, who were administered with treatment ranging from 444 to 666 MBq/kg. The children achieved an event-free survival and overall survival rates at 1-year post-therapy of 42% and 58%, respectively, without significant hematological toxicity [16].

Besides neuroblastoma, MIBG-based theragnostic has been successfully applied for the management of pediatric pheochromocytoma and paraganglioma. These are rare chromaffin cell tumors capable of synthesizing and releasing catecholamines into the blood pool. Although these tumors are quite rare in children, they can cause secondary hypertension with a peculiar clinical presentation [41]. While pheochromocytomas usually arise from the adrenal medulla, paragangliomas can occur at the head/neck, along with the sympathetic ganglia of the thoracic and abdominopelvic region [42]. Additionally, while adult pheochromocytoma and paraganglioma are sporadic, in most cases (i.e., ranging from 40 to 80%) the pediatric forms are correlated with hereditary syndromes, such as multiple endocrine neoplasia (MEN) type 2 and Von Hippel–Lindau type 2 [43]. Diagnosis of pediatric pheochromocytoma or paraganglioma can be achieved through clinical examination, laboratory tests (i.e., the dosage of catecholamine metabolites’ levels in urine), and imaging. As far as the last aspect, abdominal ultrasound examination often represents a first-line imaging approach, but its sensitivity is limited, especially in the case of small tumors.

In contrast, both computed tomography (CT) and magnetic resonance imaging (MRI) present higher values of diagnostic accuracy [44]. ^123^I-MIBG scan, especially when combined with MRI or CT, has been found characterized by specificity ranging from 95% to 100%. Furthermore, it also represents a whole-body approach; therefore, allowing the localization of multifocal sites of disease [45]. Figure 2 illustrates a clinical case in which the suspicion of pediatric pheochromocytoma has been excluded through a ^123^I-MIBG whole-body scan and combined SPECT/MRI technology.

As regards the therapeutic counterpart, it has to be highlighted that, in the case of pheochromocytoma/paraganglioma, surgery represents the treatment of choice. However, the utilization of ^131^I-MIBG has been reported in case of ineffective surgery due to difficult anatomical access (i.e., retroperitoneal localization) or in patients with diffuse metastases of the rare malignant forms. In such cases, radionuclide therapy has been exploited to obtain a complete recovery or reasonable disease control [17].

### 2.2. Somatostatin Analogs

Somatostatin (SST) is a hormone responsible for multiple functions, acting as a neurohormone, neurotransmitter, and autocrine/paracrine hormone, through its binding to somatostatin receptors (SSTR, 1-5). [46]. Since several tumors express SSTR, targeted therapy with SST analogs may be used for malignancies treatment [47]. Neuroblastoma cells may present SSTRs as well, thereby permitting positron emission tomography/computed tomography (PET/CT) imaging with somatostatin analogs and PRRT. Several radiopharmaceuticals have been developed in the diagnosis and treatment of neuroblastoma. Gallium 68 (^68^Ga)-[tetraxetan-D-Phe1, Tyr3]-octreotate (68Ga DOTATATE) and ^68^Ga-[tetraxetan-d-Phe1, Tyr3]-octreotide (^68^Ga DOTATOC) are radiolabeled somatostatin analogs used in PET/CT imaging for diagnostic purposes. A third somatostatin analog radiopharmaceutical with an affinity for SSTR 3, ^68^Ga-[tetraxetan]-1-NaI3-octreotide (^68^Ga DOTANOC), has not been widely studied in neuroblastomas, which more frequently express SSTR 2 [48]. Therefore, a theragnostic treatment based on the somatostatin analogs has been developed using ^177^Lu, ^111^In, and ^90^Y.

^68^Ga-DOTATATE can be used to image children with neuroblastoma and identify those suitable for molecular radiotherapy with^177^Lu-DOTATATE. A previous paper has shown that treatment with ^177^Lu-DOTATATE is safe and feasible in children with relapsed or primary refractory high-risk neuroblastoma [18].

In a previous paper ^68^Ga DOTATATE PET was positive in a high proportion of children with refractory neuroblastoma, correlating with SSTR 2 on immunohistochemistry, with additional disease identified compared with MIBG imaging. They performed radionuclide therapy with^111^In-DOTATATE, ^177^Lu-DOTATATE therapy, combined ^111^In-DOTATATE and ^177^Lu-DOTATATE, and combined ^177^Lu-DOTATATE and ^90^Y-DOTATATE, demonstrating the safety, feasibility and efficacy of radionuclide therapy in these pediatric patients [19].

NETs represent another crucial oncological field in which PRRT plays a crucial role. These tumors, arising from cells located in the pulmonary and gastrointestinal tract, can produce a variety of symptoms (i.e., diarrhea, cutaneous flushing, among others) due to the release of several bioactive molecules into the blood pool (i.e., the so-called “carcinoid syndrome”). NETs rarely occur both in adults and in children, although their incidence seems to be increasing in the last years [49]. Despite their rarity, NETs represent the most frequent malignancy of the gastrointestinal tract in children and adolescents [50].

It has to be underlined that both ^177^Lu and ^90^Y have been exploited for PRRT of NETs. It is worth mentioning that while ^90^Y is an almost pure beta- emitter, with a maximum energy of 2.3 MeV and tissue penetration of 11 mm, ^177^Lu presents lower energy (0.5 MeV) and a range of penetration of 2 mm. Furthermore, ^177^Lu is also characterized by a gamma co-emission that can be utilized for the in vivo imaging. Figure 3 depicts the case of an adolescent affected by an ileal NET with hepatic metastases, treated with ^177^Lu-DOTATATE.

A turning point for PRRT was represented by approval by the Food and Drug Administration (FDA) in 2018 of Lutathera^®^ (^177^Lu-DOTATATE) for the treatment of progressive neuroendocrine tumors. It consists of the fixed activity of 7.4 GBq administered in 4 cycles at an interval of 8 weeks [51].

## 3. Theragnostic Tracers in Pediatric CNS Tumor

Tumors of the central nervous system (CNS) are the second most frequent malignancy in pediatric patients. Low-grade gliomas are the most common type (40% of all CNS tumors in childhood and adolescence) followed by embryonal tumors (15%, this group includes medulloblastoma) and high-grade gliomas (11,1%) [52]. To date, surgery, radiotherapy, chemotherapy, immunotherapy, and targeted therapy are used to treat pediatric brain tumors [53,54]. Concerning the field of neuro-oncology in nuclear medicine, the use of both diagnostic [55] and theragnostic radiopharmaceuticals is growing. Molecular imaging allows a personalized approach to the patient that may provide useful information regarding tumor diagnosis and evaluation of therapeutic efficacy [3]. Nevertheless, there are few studies concerning their utilization in pediatric patients.

CNS tumors may express somatostatin receptors (SSTR), and in particular, SSTR subtype 2 is frequently present in medulloblastoma [56]. Children with CNS tumors may benefit from radionuclide therapy with radiolabeled somatostatin analogs. In a previous study, Menda et al. demonstrated the feasibility and safety of ^90^Y-DOTATOC therapy in pediatric patients with somatostatin receptor-positive tumors, including children with medulloblastoma and anaplastic astrocytoma [20]. Furthermore, ^68^Ga-Dota-Peptides PET/CT may be useful to select the candidates for targeted therapy with radiolabeled somatostatin analogs [55].

Bevacizumab (Avastin^®^) is a humanized monoclonal antibody with an antiangiogenic effect, inhibiting vascular endothelial growth factor (VEGF). It is widely used in the treatment of pediatric brain tumors [57]. PET/CT with zirconium 89 (^89^ Zr)-bevacizumab, targeting VEGF, may be helpful in the identification of pediatric patients suitable for therapy with bevacizumab. In a previous study, Jansen et al. demonstrated the practicability of ^89^ Zr-bevacizumab PET/CT in 7 children with diffuse intrinsic pontine glioma (DIPG), a childhood chemo resistant malignancy with poor prognosis originating in the pons [21]. Moreover, in a case reported by van Zanten et al., the authors found a close relationship between ^89^ Zr-bevacizumab uptake and tumoral vascular proliferation and histology in a 12-year-old patient with DIPG. This suggested the utilization of ^89^ Zr-bevacizumab PET to assess a possible therapy with Bevacizumab [22].

Regarding DIPG, 8H9 labeled with iodine 124 (^124^I) may be another promising theragnostic tracer in nuclear medicine. ^124^I-8H9 is a radiolabeled monoclonal antibody that binds B7-H3, a membrane protein expressed in DIPG and other brain tumors. This radiopharmaceutical may be used in PET, to evaluate a β-emitter radiation therapy with the same tracer or ^131^I-8H9, delivering it into the target area with an infusion technique called convection-enhanced delivery (CED), The radiopharmaceutical is infused directly through a catheter implanted into the tumoral lesion to bypass the blood-brain barrier (BBB) [23,58].

Optic pathway glioma (OPG) is a typical childhood tumor, and it is the most frequent neoplasm of the anterior visual pathway. It is usually associated with neurofibromatosis 1. The therapy comprises radiotherapy, chemotherapy, surgery, or conservative follow up (in benign lesions) [59]. The well-known expression of gastrin-releasing peptide receptor (GRPR, a bombesin family receptor) in several tumors [60] and OPG, opens the door to attractive theragnostic applications. The role of GRPR targeted PET has been evaluated through the utilization of ^68^Ga-NOTA-Aca-BBN(7-14). This radiopharmaceutical targets GRPR specifically, in pediatric patients with OPG [24]. ^68^Ga-NOTA-Aca-BBN(7-14) PET showed promising results in terms of feasibility in children and correlation with GRPR expression in optic pathway glioma, thereby offering potential guidance for GRPR-targeted therapy in these patients.

## 4. Tracers for Radioimmunotherapy

A further innovative field is radioimmunotherapy that aims to improve the management of pediatric patients using the synergic effects between immunotherapy and radiation. The most used radioimmunotherapy tracer is ^90^Y-radiolabeled ibritumomab tiuxetan, Zevalin^®^, routinely administered for treatment purposes in patients with rituximab-relapsed or -refractory CD20+ follicular B- cell non-Hodgkin’s lymphoma (NHL) since 2004 [61]. ^90^Y is a high-energy beta-emitting radioisotope with an X90 (a measure of the radius in which the isotope deposits 90%).

Ibritumomab tiuxetan is composed of the monoclonal antibody ibritumomab, from which the well-known chimeric monoclonal antibody rituximab (Mabthera^®^) is obtained, as well as tiuxetan, the chelator molecule linked to the radionuclide [61]. Therefore, Zevalin^®^ allows to target CD20+ cells along with bystander cells within 5 mm with a half-life of 64 h. This allows for targeted therapy to areas containing CD20+ cells in contrast to indiscriminate total body irradiation [62]. The efficacy of radioimmunotherapy with Zevalin^®^ has been well-demonstrated, with low toxicity for healthy tissues. [6]

A particular aspect of radioimmunotherapy is the possible synergic role of unlabeled drugs. Pre targeting with unlabeled chimeric monoclonal antibodies (Mab) (=preload) as part of a treatment with Zevalin^®^ leads to a more favorable biodistribution of the subsequently injected ^90^Y ibritumomab tiuxetan, avoiding its possible binding with healthy sites (such as normal B-cells, spleen). Moreover, pretargeting with unlabeled Mab allows a more effective and homogenous penetration into the tumor by the radiolabeled antibody [61]. Ibritumomab tiuxetan can be conjugated to Indium-111 (^111^In) to form ^111^In ibritumomab tiuxetan (^111^In-Zevalin) used for scanning and dosimetry or ^90^Y to form ^90^Y ibritumomab tiuxetan (^90^Y-Zevalin) for therapy [62]. Future developments in ^90^Y-radiolabeled ibritumomab tiutumourin NHL may concern its pre-therapeutic implications in the management of the patient.

To date, pre-therapeutic imaging with ^111^In-ibritumomab tiuxetan is mandatory in Switzerland and the United States but not in the European Union. Due to the lack of reliable evidence about the usefulness of pre-therapeutic dosimetry in Zevalin^®^ therapy, the imaging is performed to add a further safety measure before radioimmunotherapy administration confirming the foreseen biodistribution and not for dosimetric aims [61]. Nevertheless, the pretherapeutic assessment with ^111^In-labeled compound imaging is mandatory when Zevalin^®^ therapy is performed for experimental and investigational purposes [61], allowing a personalized and a theranostic approach as well.

The protocol of radioimmunotherapy with Zevalin^®^ is well established, but further studies may explore different indications in children’s oncology and the dosimetric field.

As concerns the application of radioimmunotherapy in pediatric brain tumors, few studies have been conducted. The monoclonal antibody 3F8 binds the disialoganglioside GD2, expressed on the plasma membrane of medulloblastoma cells. A previous paper tested the intraventricular ^131^I-labeled 3F8 in patients with medulloblastoma. High-risk and recurrent medulloblastoma is associated with significant mortality. Pre-therapeutic and therapeutic assessment authors evaluated the efficacy, toxicity, and dosimetry of compartmental radioimmunotherapy and suggested that ^131^I-labelled3F8 may have clinical utility in maintaining remission in high-risk or recurrent medulloblastoma, with favorable dosimetry. Dosimetry to the cerebrospinal fluid (CSF) in the ventricles and within the thecal sac was determined by region-of-interest analysis ^124^I-3F8/PET imaging or ^131^I-3F8/ SPECT gamma camera imaging [25]. ^131^I-8H9, binding the membrane protein B7-H3, may present another attractive radiolabeled antibody for targeted radioimmunotherapy in children with brain tumors. In three pediatric patients with embryonal tumors, radioimmunotherapy with ^131^I- 8H9 seemed to be safe, and SPECT-based dosimetry showed an advantageous therapeutic index [26]. Moreover, it has demonstrated a minimal risk of radionecrosis in children treated with intraventricular radioimmunotherapy with radiolabeled antibodies ^131^I-8H9 and ^131^I-3F8. In a new retrospective study, Kramer et al. investigated the incidence of radionecrosis in pediatric patients with medulloblastoma or metastatic CNS neuroblastoma treated with both external beam radiotherapy and intraventricular radioimmunotherapy. The risk of radionecrosis in these patients was shallow (~1%), demonstrating the safety of radiation treatment with radiolabeled antibodies. [27].

In a recent study, Souweidane et al. showed the safety of ^124^I-8H9 (^124^I-omburtamab) radioimmunotherapy performed with CED infusion in 37 children with DIPG. Dosimetry of the radiopharmaceutical was determined by PET imaging. CED infusion allowed high intra-tumoral dosing with a minimal level of systemic radiation exposure [28]. The same research group demonstrated similar results in a previous study [23]. Nevertheless, the main drawback of CED infusion in children with DIPG may be represented by subsequent but self-limited volumetric alterations in the pons [29].

## 5. Discussion

The role of MIBG in neuroblastoma is well established. The initial evidence of the effectiveness of MIBG neuroblastoma dates back to 1986 [63]. Nevertheless, neuroblastoma is the most common extracranial solid tumor of childhood, with an aggressive nature. Even if MIBG therapy protocol is well established in neuroblastoma management, actual developments about theragnostics with MIBG are focused mainly on the dosimetry and toxicity, in some cases combined with other treatments, to improve the effectiveness and to reduce collateral effects of the treatment.

A previous multicentric study aimed to determine the response rate, survival, and toxicity of tandem infusions of high activity ^131^I MIBG in children with relapsed/refractory neuroblastoma. The authors described the feasibility of high-activity ^131^I MIBG under strict radiation protection precautions. In all patients, a pre-therapeutic ^123^I MIBG has been performed [30]. Moreover, a previous paper described the combination with vorinostat at 180 mg/m(2)/dose and ^131^I MIBG [31]. The pre-therapeutic scan leads to the eligibility of patients based on a personalized approach. Moreover, the pre-therapeutic dosimetry has a predictive role on possible collateral effects of MIBG theragnostic therapy in children, mainly due to the variable degree of myelosuppression and, therefore, influencing and determining patient management.

Moreover, in the somatostatin analogs field, the pre-therapeutic assessment enables noninvasive quantitation for individualized dosimetry estimates for tumor burden, whole body, and critical organs for a subsequent theragnostic approach [64]. In this regard, it has to be underlined that dosimetric assessment represents a crucial issue for PRRT in children, particularly as far as it concerns nephrotoxicity in the case of therapy with ^90^Y. Despite renal protection provided by the co-infusion of cationic amino acids, the deterioration of renal function (grade 4–5 toxicity) has been reported in 9% of the treated subjects [65]. In such cases, personalized dosimetry should be performed to achieve a patient’s tailored PRRT, ensuring not to exceed the limiting dose of 23 Gy to kidneys. Menda et al. developed an interesting approach based on the small positron emission of ^90^Y so that they performed both ^90^Y-DOTATOC PET/CT and bremsstrahlung SPECT/CT for dosimetric calculation. In a recently published report including two children and two young adults, personalized dosimetry was found to have a significant impact on patient management and the administered activities in the subsequent cycles of PRRT [66].

Although nephrotoxicity has rarely been reported when ^177^Lu is used, the recently approved (by the FDA) Lutathera^®^, which is based on the administration of fixed doses, might have limitations in the pediatric population, since the adsorbed dose in bone marrow and kidneys may present a significant variability among patients. On this path, it is worth mentioning the ongoing phase I/II clinical trial (https://clinicaltrials.gov/ct2/show/NCT03923257). It is aimed at assessing the effectiveness and safety of a dosimetry-guided PRRT with ^177^Lu-DOTATATE in children with relapsed/refractory neuroendocrine tumors (pheochromocytoma, paraganglioma, gastrointestinal NETs) or neuroblastoma. The results of the aforementioned clinical trial will be of great importance for further defining the role of PRRT in the pediatric setting.

The incremental diagnostic information and convenient imaging protocol suggest ^68^Ga-DOTATATE PET/CT could become the preferred molecular imaging technique for pediatric neuroblastoma patients. Further prospective studies may include ^68^Ga-DOTATATE PET/CT for primary staging, to measure if theranostic by somatostatin analogs might be an alternative to ^131^I-MIBG in first-line treatment of neuroblastoma. Moreover, compared with a fixed-dose peptide receptor radionuclide therapy protocol, an adjusted-dose regimen tailored to tumor burden, body habitus, and renal function may allow a more significant radiation dose to individual lesions without substantially adding to toxicity in healthy tissues [64].

Another innovative theragnostic approach to neuroblastoma, also exploiting the overexpression of somatostatin receptors, is represented by the applications of copper (Cu) radioisotopes (^64^Cu/^67^Cu). The ^64^Cu radionuclide constitutes a real theragnostic agent since it can be used, at different dosages, for diagnostic or imaging [67]. On the one hand, the relatively long half-life (i.e., 12.7 hours) and positron emission make this radionuclide particularly suitable for PET imaging, on the other hand, its decay via beta particles emission (39%, 0.573 MeV) also allows the application of ^64^Cu for therapeutic purposes. A recently synthesized compound, capable of binding to tumors overexpressing somatostatin receptors, namely ^64^Cu-MeCOSar-Tyr^3^-octreotate (^64^Cu-SARTATE), was found useful for imaging, dosimetry, and PRRT of NETs [32]. Furthermore, in the radiolabeling of SARTATE, the radionuclide ^64^Cu can be easily substituted with the isotope ^67^Cu, which emits beta particles (Emax = 0.56 MeV) and also photons suitable for imaging. New data for the potential use of the theragnostic couple ^64^Cu/^67^Cu-SARTATE in neuroblastoma will be provided by an ongoing clinical trial (https://clinicaltrials.gov/ct2/show/NCT04023331).

Nevertheless, considering the high mortality rate of neuroblastoma, new targeted treatment options are urgently needed. 18-(p-[127I] iodophenyl) octadecyl phosphocholine (CLR1404) is a novel, broadly tumor-targeted small molecule drug with highly selective tumor uptake tested in human cells and in animal models with encouraging results [68].

^131^I-CLR1404 has the potential to become a tumor-targeted radiotherapeutic drug with broad applicability in pediatric oncology [2]. Radiolabeled guanidine (R)-(-)-5-iodo-3’-*O*-[2-(ε-guanidinohexanoyl)-2-phenylacetyl]-2’-deoxyuridine (GPAID), targeting DNA of neuroblastoma cells, maybe a promising and attractive theragnostic tool as well, allowing an Auger-electron therapy with the radioisotope ^125^I [33].

A further innovative field is radioimmunotherapy that aims to improve the management of pediatric patients using the synergic effects between immunotherapy and radiation. This approach can potentially be used in several pathologies, based on novel immunotherapy developments. Zevalin^®^ is the most used example of radioimmunotherapy, and it is used for decades. New developments about Zevalin^®^ treatment include dosimetry assessment, further indications, and combination with different treatments [61].

As regards the utilization of theragnostic radiopharmaceuticals in children with brain tumors, pediatric patients with recurrent medulloblastoma, one of the most aggressive childhood tumors, may benefit from somatostatin receptor imaging to allow a possible radionuclide therapy with radiolabeled somatostatin analogs. Further studies are needed, but ^90^Y-DOTATOC therapy may be promising. Moreover, given the lack of γ radiation emission of ^90^Y, children isolation after therapy is not required, thereby leading to a better quality of life during treatment in comparison to other radiopharmaceuticals (such as tracers labeled with ^131^I). Regarding intraventricular radioimmunotherapy in patients with medulloblastoma, groundbreaking radiolabeled antibodies such as ^131^I-8HD, and ^131^I-3F8 may be effective. However, this is only in the absence of significant obstructions of CSF. If not, a surgical procedure before radioimmunotherapy should be considered to resolve the obstruction [69].

DIPGs are aggressive tumors with poor prognosis and a median overall survival of less than one year. Treatment of DIPG is challenging. The anatomical localization makes surgery problematic, and chemotherapy has no effect, given the difficulty of drugs to cross the BBB (often intact in DIPG) [70]. That is why the potential role of nanoparticles (such as gold nanoparticles, liposomes, and dendrimers) in the treatment is noteworthy, to deliver chemotherapies into DIPGs crossing the BBB [71]. ^124^I/^131^I-8HD radioimmunotherapy with CED infusion, bypassing the BBB, maybe promising in the treatment of these patients as well. Moreover, a selection of children suitable for bevacizumab therapy may be performed with ^89^ Zr-bevacizumab PET, predicting the possible advantage of antiangiogenetic treatment.

Concerning malignant OPGs, surgery is often complicated, and treatment options are usually radiotherapy and chemotherapy [72]. The potential role of GRPR targeted therapy may add a new exciting weapon in the challenging treatment of these young patients. In this regard, ^68^Ga-NOTA-Aca-BBN(7-14) PET could be useful in the selection of patients with a high chance of benefit from GRPR targeted therapy.

Further approaches in pediatric oncology include the implantation of seeds labeled with a therapeutic-emitting radionuclide. Yao et al. demonstrated the feasibility, efficacy, and potential use of permanent interstitial ^125^I seed implantation in children with recurrent or metastatic soft tissue sarcoma [34].

## 6. Conclusions

Concerning the personalized therapy in children’s oncology, this review has underlined the impact and the importance of contributions of nuclear medicine and its future perspectives. Further studies are needed, but the growing use of theragnostic radiopharmaceuticals in pediatric tumors may help to consolidate the critical role of personalized therapies in children, integrating diagnosis and therapy efficiently, and selecting patients with the most excellent chance of benefit from specific treatments. Moreover, the groundbreaking developments in the field of pediatric radioimmunotherapy could often support a theragnostic approach, adding new weapons and new possibilities to conventional therapies as well. Considering the aggressivity and poor prognosis of typical childhood malignancies such as neuroblastoma or CNS tumors, new targeted therapies are needed. Theragnostic radiopharmaceuticals and radioimmunotherapy may efficiently satisfy these requests, allowing an intra-personal approach.

## Figures and Tables

**Figure 1 ijms-21-03849-f001:**
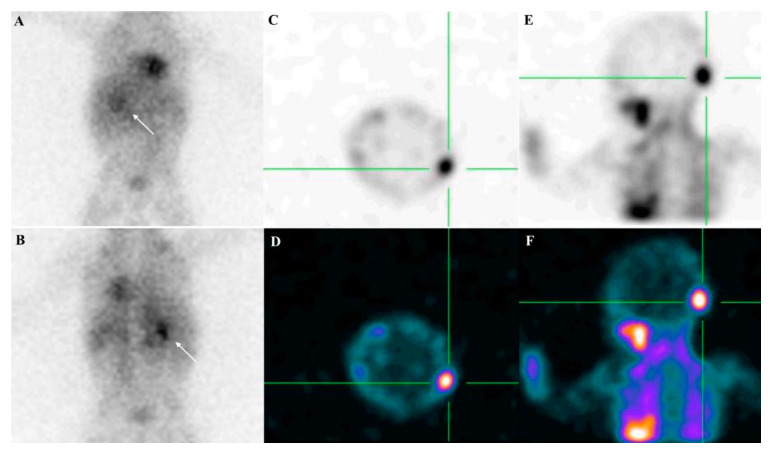
Pre-therapeutic staging in a 4-year-old child with neuroblastoma, performed with ^123^I MIBG scintigraphy/ single-photon emission computed tomography (SPECT). Planar images (**A**, anterior; **B**, posterior) show high uptake of the tracer in the right adrenal gland (white arrow), corresponding to the primary site of the tumor. Moreover, SPECT images of the skull (**C**–**F**) demonstrate the presence of metastasis in the occipital bone. Figures E and F display the physiological activities in the salivary glands and in the heart.

**Figure 2 ijms-21-03849-f002:**
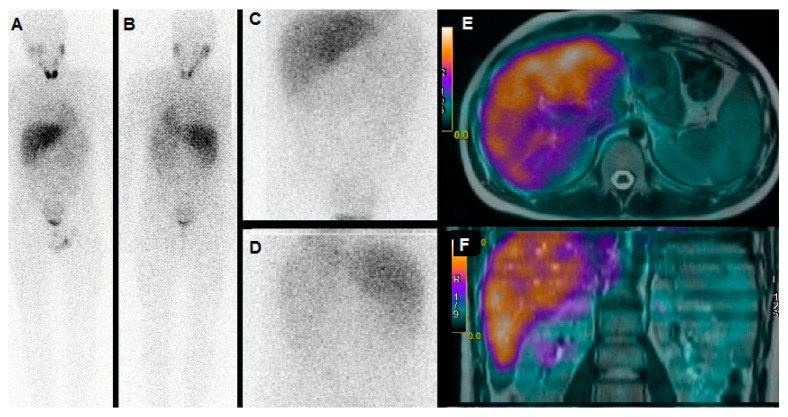
A 13-year-old boy suffering from hypertension, with evidence of slightly increased levels of catecholamine metabolites in urine. ^123^I-MIBG scintigraphy was carried out for the suspicion of pheochromocytoma. Whole Body scan in anterior (**A**) and posterior (**B**), abdominal planar (**C**, anterior; **D**, posterior) did not show any area of abnormal tracer uptake. Co-registered corresponding axial (**E**) and coronal (**F**) SPECT/MRI images confirmed the absence of pathological hyperactive masses in both adrenal glands.

**Figure 3 ijms-21-03849-f003:**
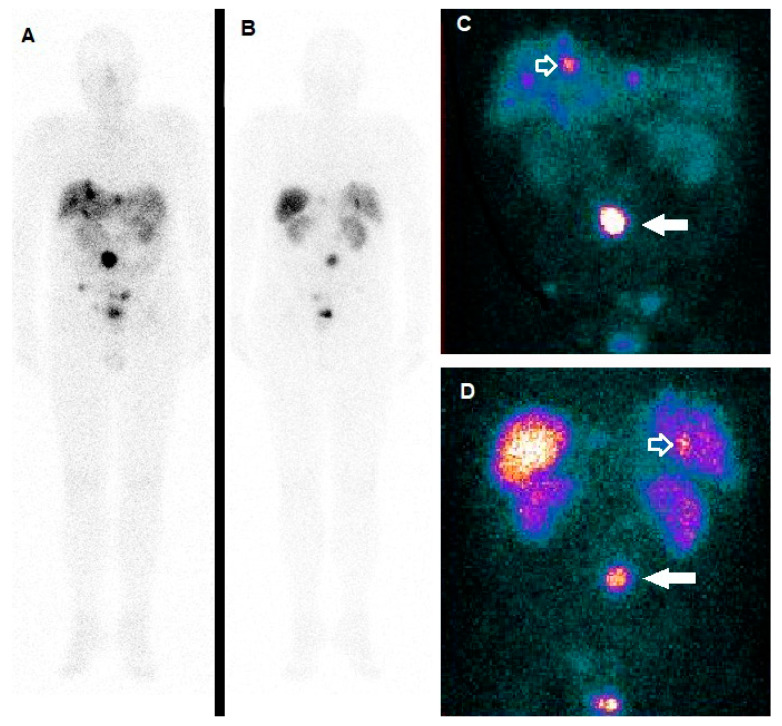
A 17-year-old male with mental retardation due to perinatal hypoxia, suffering from carcinoid syndrome due to ileal NET with hepatic metastases. Since symptomatology persisted despite somatostatin analog therapy and CT gave evidence of tumor progression, PRRT with 177Lu-DOTATATE was considered. Whole Body scan in anterior (**A**) and posterior (**B**) view, as well as the planar (**C**, anterior; **D**, posterior), acquired with gamma camera centered on the ^177^Lu photopeak at 113 KeV, well-demonstrated tracer incorporation both in the tumor (**C**,**D**, white arrow) and in the liver lesions (**C**,**D**, white-bordered arrow). Three further focuses of increased accumulation of the radiopharmaceutical are visible under the primary tumor in the pelvic region, especially evident in the anterior view (**C**), to be referred to peritoneal carcinomatosis.

**Table 1 ijms-21-03849-t001:** Summary of the most relevant studies focused on theragnostics and radioimmunotherapy cited in the paper.

Authors	Year	Radiopharmaceutical	Setting	Population	Comments
[14] Baum R.P. et al.	2018	^90^Y -based peptide receptor radionuclide therapy (PRRT); ^177^Lu-based PRRT	PRRT	1048 adult patients with somatostatin receptor-expressing neuroendocrine neoplasms	PRRT is adequate and overall survival is favorable in patients with neuroendocrine tumors by highly sensitive ^68^Ga somatostatin receptor positron emission tomography/computed tomography (PET/CT).
[15] Suh J.K. et al.	2020	^131^I-Metaiodobenzylguanidin (^131^I-MIBG)	^131^I-MIBG therapy in neuroblastoma	18 patients with neuroblastoma	Feasibility of ^131^I MIBG therapy in combination with high dose chemotherapy and autologous stem transplantation.
[16] Kayano D. et al.	2020	^131^I-MIBG	^131^I-MIBG therapy in neuroblastoma	20 patients with neuroblastoma	Safety and feasibility of high dose ^131^I-MIBG therapy in pediatric patients with neuroblastoma.
[17] Hasse-Lazar K. et al.	2008	^131^I-MIBG	^131^I-MIBG therapy in pheochromocytoma	Three patients with pheochromocytoma	Effectiveness of ^131^I-MIBG treatment in pediatric patients with pheochromocytoma.
[18] Gains J.E. et al.	2011	^177^Lu-[tetraxetan-d-Phe1, Tyr3]-octreotate (^177^Lu-DOTATATE)	PRRT in neuroblastoma	Eight pediatric patients with high-risk neuroblastoma	^68^Ga- DOTATATE can be used to image children with neuroblastoma and identify those suitable for molecular radiotherapy with ^177^Lu-DOTATATE. Safe and feasible in children with relapsed or primary refractory high-risk neuroblastoma.
[19] Kong G. et al.	2016	^177^Lu-DOTATATE; ^111^In-DOTATATE; ^90^Y-DOTATATE	PRRT in neuroblastoma	Eight pediatric patients with high-risk neuroblastoma	Safety and feasibility of PRRT in children with refractory neuroblastoma.
[20] Menda Y. et al.	2010	^90^Y-68Ga-[tetraxetan-d-Phe1, Tyr3]-octreotide (^90^Y-DOTATOC)	PRRT in neuroblastoma, embryonal and astrocytic brain tumors, paraganglioma, neuroendocrine tumors	17 pediatric and young adults (age, 2–24 years) patients with refractory solid tumors	Safety of ^90^Y-DOTATOC therapy in pediatric somatostatin receptor-positive malignancies.
[21] Jansen M. H. et al.	2017	^89^Zr- bevacizumab	Pre-therapy distribution assessment in diffuse intrinsic pontine glioma (DIPG)	Seven patients (4 boys; 6–17 years old) with DIPG	Tumor ^89^Zr-bevacizumab accumulation assessed by PET scanning may help in the selection of patients with the most excellent chance.
[22] Veldhuijzen van Zanten S. E.M et al.	2018	^89^Zr-bevacizumab	In vivo and ex vivo measure of metastasis samples	One patient (12 years old) with DIPG	In vivo ^89^Zr-bevacizumab PET serves to identify heterogeneous uptake between tumor lesions.
[23] Souweidane M.M. et al.	2018	^124^I-8H9	Radioimmunotherapy and pre-therapy distribution assessment in DIPG	28 pediatric patients (3–21 years old) with DIPG	PET-based dosimetry of the radiolabeled antibody ^124^I-8H9 validated the principle of using convection-enhanced delivery in the brain to achieve high intra-lesional dosing with negligible systemic exposure.
[24] Zhang J. et al.	2019	^68^Ga-NOTA-Aca-BBN(7-14)	Radioimmunotherapy and pre-therapy distribution assessment in optic glioma	Eight pediatric patients (5–14 years old) with suspicion of optic pathway glioma	Gastrin-releasing peptide receptor(GRPR)-targeted PET has the potential to provide imaging guidance. for further GRPR-targeted therapy in patients with Optic pathway glioma
[25] Kramer K. et al.	2018	^131^I-labeled 3F8	Phase II clinical trial for intraventricular compartmental radioimmunotherapy in medulloblastoma	43 pediatric patients with medulloblastoma	Safety and potential clinical applications of radioimmunotherapy with ^131^I-3F8 in patients with medulloblastoma.
[26] Bailey K. et al.	2019	^131^I-omburtamab (8H9)	Intraventricular compartmental radioimmunotherapy(cRIT) in embryonal tumor with multilayered rosettes	Three pediatric patients with embryonal tumor with multilayered rosettes	^131^I-omburtamab appears safe with a favorable dosimetry therapeutic index.
[27] Kramer K. et al.	2015	^131^I-3F8; ^131^I-8H9	cRIT in brain primary and secondary lesions	94 pediatric patients with metastatic CNS neuroblastoma and medulloblastoma	Administration of cRIT may safely proceed in patients treated with conventional radiotherapy without appearing to increase the risk of radionecrosis.
[28] Souweidane M.M. et al.	2019	^124^I-8H9	Convection-enhanced delivery (CED) and pre-therapy distribution assessment in pontine glioma	37 pediatric patients with diffuse intrinsic pontine glioma	CED in the brain stem of children with DIPG who were previously irradiated is a safe therapeutic strategy.
[29] Bander E.D. et al.	2020		CED and treatment-related volumetric alterations	23 pediatric patients with diffuse intrinsic pontine glioma	CED infusion into the brainstem correlates with immediate but self-limited deformation changes in the pons.
[30] Cougnenc O. et al.	2017	^131^I-MIBG	Theragnostics in neuroblastoma	15 pediatric patients with neuroblastoma	Under strict radiation protection precautions, this study shows the feasibility of high-activity ^131^I -MIBG therapy in France.
[31] DuBois S.G. et al.	2015	^131^I-MIBG	Theragnostic and vorinostat combination in neuroblastoma	27 children and Young adults	Vorinostat at 180 mg/m(2)/dose is tolerable with 18 mCi/kg MIBG.
[32] Hicks R.J. et al.	2019	^64^Cu-MeCOSar-Tyr3-octreotate (^64^Cu-SARTATE)	Somatostatin receptor imaging in neuroendocrine tumors	Ten patients with neuroendocrine tumors	Safety and effectiveness of ^64^Cu-SARTATE PET/CT imaging in patients with neuroendocrine tumors. Potential applications for ^67^Cu-SARTATE therapy.
[33] Kortylewicz Z.P. et al.	2020	(R)-(-)-5-[^125^I]iodo-3’-*O*-[2-(ε-guanidinohexanoyl)-2-phenylacetyl]-2’-deoxyuridine (9, GPAID)	Theragnosticsin neuroblastoma in vivo and in vitro assessment in mice models	Mice models	The chemical structure accommodates therapeutic, as well as diagnostic radionuclides. Biological properties of GPAID suggest its significant potential as a novel theragnostic tracer for the management of neuroblastoma
[34] Yao L. et al.	2015	^125^I	Interstitial ^125^ Iseed-implantation in sarcoma	Ten patients with soft tissue sarcoma	Potential use and feasibility of interstitial ^125^I seed implantation therapy in pediatric patients with metastatic or recurrent sarcoma.

**Table 2 ijms-21-03849-t002:** Summary of the different radiopharmaceuticals used for pediatric oncology in theragnostics and radioimmunotherapy, with their molecular target.

Radiopharmaceutical	Molecular Target	Nuclear Medicine Applications
^123^I/^131^I MIBG	Norepinephrine transporters	Theragnostic applications in children with neuroblastoma.
Radiolabeled somatostatin analogs	Somatostatin receptors	Theragnostic applications in children with neuroblastoma and brain tumors (especially medulloblastoma).
^89^Zr-bevacizumab	Vascular endothelial growth factor (VEGF)	In pediatric patients with DIPG, ^89^Zr-bevacizumab PET may select patients with possible benefit from antiangiogenetic therapy.
^124^I/^131^I-8H9	B7-H3	Theragnostic and radioimmunotherapy applications in children with brain tumors.
^131^I-3F8	GD2	Radioimmunotherapy in pediatric patients with medulloblastoma.
^68^Ga-NOTA-Aca-BBN	Gastrin-releasing peptide receptor (GRPR)	In children with optic pathway glioma, ^68^Ga-NOTA-Aca-BBN PET may select patients with possible benefit from GRPR targeted therapy.
^64^Cu/^67^Cu-SARTATE	Somatostatin receptors	In children with neuroblastoma, it is providing the opportunity for personalized dosimetry.
^90^Y-ibritumomab tiuxetan	CD20	This radiolabeled antibody allows radioimmunotherapy applications in patients with low-grade Non-Hodgkin’s lymphoma (relapsed or refractory).

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
