# Peer review of "Theragnostic Aspects and Radioimmunotherapy in Pediatric Tumors"

_ijms, 2020, doi:10.3390/ijms21113849_

Round 1

Reviewer 1 Report

This manuscript reviewed the potential role of theragnostic imaging in pediatric oncology, especially in patients with aggressive malignancies such as neuroblastoma and brain tumors, to select patients with the highest chance of benefit from personalized therapy. Several tracers were considered, including MIBG and SST analogues. Both diagnostic and therapeutic (PRRT) radiopharmaceuticals were discussed.

Moreover, the most promising applications of radioimmunotherapy in children’s oncology were assessed. The authors concluded that theragnostic imaging and radioimmunotherapy in pediatric oncology are useful in current applications of nuclear medicine, allowing an intra-personal approach and adding new weapons to conventional therapies.

The review is comprehensive and very-written and it is of interest to the journal. I have only very few minor comments.

Line 76, change further iodine 
to other iodine

Line 91 (and throughout the text): use theragnostic, not theranostic

Line 224 “almost beta- emitter,“. I think the authors here mean „almost pure“

Figure 1: the authors should very briefly describe the two other hot areas in Figure 1E and 1F (gland? heart?). Just say something like: “Displayed are also activities in … and in …”.

Figure 3: in the anterior view there at least 2 area of focal uptake under the tumor. Are these physiological uptake or peritoneal sites? Just provide a brief explanation in the legend.

Author Response

Response to Reviewer 1 Comments

Point 1: Line 76, change further iodine to other iodine

Response 1: As kindly suggested, we have changed “further iodine” to “other iodine”(line 76, in red).

Point 2: Line 91 (and throughout the text): use theragnostic, not theranostic

Response 2: As kindly suggested, we have changed “theranostic” to “theragnostic” and “theranostics” to “theragnostics” (throughout the text, the title, the abstract, the keywords and the tables as well, in red).

Point 3: Line 224 “almost beta- emitter,“. I think the authors here mean „almost pure“

Response 3: yes, we mean “almost pure “ (line 224, we have inserted “pure” , in red).

Point 4: Figure 1: the authors should very briefly describe the two other hot areas in Figure 1E and 1F (gland? heart?). Just say something like: “Displayed are also activities in … and in …”

Response 4: As kindly suggested, we have described the two hot areas in figures 1E and 1F (in the description of figure 1 we have inserted the phrase “in figures E and F, displayed are also physiological activities in the salivary glands and in the heart”, in red).

Point 5: Figure 3: in the anterior view there at least 2 area of focal uptake under the tumor. Are these physiological uptake or peritoneal sites? Just provide a brief explanation in the legend.

Response 5:  As kindly suggested, we have described these areas of malignant uptake of the tracer under the tumor (in the description of figure 3 we have inserted the phrase “Three further focuses of increased accumulation of the radiopharmaceutical are visible under the primary tumor in the pelvic region, especially evident in the anterior view (C), to be referred to peritoneal carcinomatosis.”, in red). Previously, we have forgotten to describe these areas.

Reviewer 2 Report

Metaiodobenzylguanidine has been used for diagnosis and treatment of resistant/relapsed and newly diagnosed neuroblastoma patients for the last 40 years. In this review, to describe the use of MIBG for treatment of neuroblastoma, only four papers have been taken into consideration, but I believe they don’t cover the whole issue. I think that other papers should be mentioned as the one by Matthay K.K. et al (J Clin Oncol. 1998, 16:229-236), in which a correlation between MIBG treatment dose per body weight and hematological toxicity is reported in patients with relapsed or resistant neuroblastoma. Is also worth including a study by Mastrangelo S. et al. (British Journal of Cancer. 2001, 84:460-464), where 16 relapsed and resistant neuroblastoma patients have been treated with MIBG combined to multiple chemotherapy drugs in a 10-day course. In this paper MIBG scans were performed at start and after chemotherapy, just before therapeutic MIBG infusion at day 10, showing no reduction of MIBG uptake after chemotherapy. To better evaluate tumor real extension scintigraphy was performed weekly after MIBG treatment, showing more tumor lesions than seen on staging scan. The MIBG chemotherapy combination treatment was used by the same authors for newly diagnosed neuroblastoma patients (Mastrangelo S. et al. Pediatric Blood and Cancer. 2011, 56:1032-1040). Treatment was feasible, no treatment related deaths were reported and hematological toxicity was acceptable even with extensively infiltrated bone marrow, while early responses, evaluated by MIBG scan repeated at about 40 days, were very good and correlated to the administered MIBG dose per body weight.

Minor errors to be corrected:

1) Capital letters are present by mistake in many words throughout the text and also in tables.

2) Some punctuation marks are inappropriately inserted in the text.

3) Acronyms should be always reported only the first time they are present and then used throughout text and tables. Authors are invited to check all acronyms used and to correct them.

Line 303…it is not specified what “Mab” is

4) English needs some improvement

The following words are incorrectly written in the manuscript and need to be changed:

Manuscript:

Line 51 …. the word “develepments” should be “developments”

Line 110 should be “…form the sympathetic nervous system”

Line 111 at the beginning of the line should be “Iodine -labeled MIBG is a functional agent…”

Line 115 the word “neoplasias” should be “neoplasia”

Line 131 the word “children” should be “child”

Line 141 the word “too” should be “to”

Line 142 the words “extraordinary evidence” should be “extremely rare evidence”

Line 187 the word “metastatization for” should be “metastates of”

Line 199 the word “neuroblastomas” should be “neuroblastoma”

Line 302 the word “unlabelled” should be “unlabeled” (this misspelled word is repeated in line 302)

Line 415 the word “radiolabelled” should be “radiolabeled” (this misspelled word is repeated may times)

Table 1:

In legend the word “theranostic should be “theranostics”

In “reference 64” the word “mousse” should be “mice”

Figure 1

In legend the word “children” should be “child”

Author Response

Response to Reviewer 2 Comments

Point 1: Metaiodobenzylguanidine has been used for diagnosis and treatment of resistant/relapsed and newly diagnosed neuroblastoma patients for the last 40 years. In this review, to describe the use of MIBG for treatment of neuroblastoma, only four papers have been taken into consideration, but I believe they don’t cover the whole issue. I think that other papers should be mentioned as the one by Matthay K.K. et al (J Clin Oncol. 1998, 16:229-236), in which a correlation between MIBG treatment dose per body weight and hematological toxicity is reported in patients with relapsed or resistant neuroblastoma. Is also worth including a study by Mastrangelo S. et al. (British Journal of Cancer. 2001, 84:460-464), where 16 relapsed and resistant neuroblastoma patients have been treated with MIBG combined to multiple chemotherapy drugs in a 10-day course. In this paper MIBG scans were performed at start and after chemotherapy, just before therapeutic MIBG infusion at day 10, showing no reduction of MIBG uptake after chemotherapy. To better evaluate tumor real extension scintigraphy was performed weekly after MIBG treatment, showing more tumor lesions than seen on staging scan. The MIBG chemotherapy combination treatment was used by the same authors for newly diagnosed neuroblastoma patients (Mastrangelo S. et al. Pediatric Blood and Cancer. 2011, 56:1032-1040). Treatment was feasible, no treatment related deaths were reported and hematological toxicity was acceptable even with extensively infiltrated bone marrow, while early responses, evaluated by MIBG scan repeated at about 40 days, were very good and correlated to the administered MIBG dose per body weight.

Response 1: As kindly suggested, we mentioned:

  • The article of Matthay K.K. et al (lines 148-151, we have inserted “In this regard, a correlation between 131I-MIBG treatment dose per body weight and hematological toxicity has been reported in a dose escalation study conducted by Matthay et al. involving 30 patients with neuroblastoma [18]”, in red).
  • The articles of Mastrangelo S et al (lines 156-164, we have inserted “As early as in 2001, Mastrangelo et al. evaluated efficacy and safety of the combined therapy with 131I-MIBG and chemotherapy in a 10-day course in 16 children with resistant and relapsed neuroblastoma [19]: the majority of patients (12 of 16) showed a partial response and the only toxicity demonstrated was hematological (neutropenia and thrombocytopenia). The same authors showed the efficacy and the feasibility of 131I-MIBG chemotherapy combination treatment in 13 pediatric patients with newly diagnosed neuroblastoma as well [20]: 40 days after the beginning of combined therapy, the responses were excellent in the majority of cases (2 children showed a complete response, 6  children had very good partial responses) and hematological toxicity was acceptable even with extensively infiltrated bone marrow.” , in red). 

Point 2: Capital letters are present by mistake in many words throughout the text and also in tables.

Response 2: As kindly suggested we have changed:

  • “Children” to “children” (line 77, in red).
  • “Peptide” to “peptide” (line 87, in red).
  • “Radiopharmaceuticals” to “radiopharmaceuticals” (line 96, in red).
  • “Pre-therapy” to “pre-therapy” (table 1, study [46], in red).
  • “Pre-therapy” to “pre-therapy” (table 1, study [50], in red).
  • “Medulloblastoma” to “medulloblastoma” (table 1, study [53], in red).
  • “Embryonal Tumor With Multilayered Rosettes” to “embryonal tumor with multilayered rosettes” (table 1, study [54]).
  • “Omburtamab” to “omburtamab” (table 1, study [54])
  • “Pre-therapy” to “pre-therapy” (table 1, study [56]).
  • “Vorinostat” to “vorinostat” (table 1, study [60]).
  • “Seed-Implantation” to “seed-implantation” (table 1, study [72]).
  • “Neuroblastoma” to “neuroblastoma” (table 2, MIBG paragraph)
  • “Neuroblastoma” to “neuroblastoma” (table 2, Radiolabelled somatostatin analogs paragraph)
  • “Medulloblastoma” to “medulloblastoma” (table 2, Radiolabeled somatostatin analogs paragraph)
  • “Bevacizumab” in “bevacizumab” (table 2, 89Zr-Bevacizumab paragraph).
  • “Vascular Endothelial Growth Factor” to “Vascular endothelial growth factor” (table 2, 89Zr-Bevacizumab paragraph).
  • “Gastrin-Releasing Peptide Receptor” to “Gastrin-releasing peptide receptor” (table 2, 68Ga-NOTA-Aca-BBN paragraph).
  • “Optic Pathway Glioma” to “optic pathway glioma” (table 2, 68Ga-NOTA-Aca-BBN paragraph).
  • “Neuroblastoma” to “neuroblastoma” (line 122, in red).
  • “Tumor” to “tumor” (line 127, in red).
  • “Hypothyroidism” to “hypothyroidism” (line 147, in red).
  • “Persistent” to “persistent” (line 147, in red).
  • “Pre-therapeutic” to “pre-therapeutic” (line 151, in red).
  • “Vascular Endothelial Growth Factor” to “Vascular endothelial growth factor” (line 282, in red).
  • “Bevacizumab” in “bevacizumab” (line 285, in red).
  • “Neurofibromatosis” in “neurofibromatosis” (line 300, in red).
  • “Lack” to “lack” (line 336, in red).
  • “Omburtamab” to “omburtamab” (line 366 in red)
  • “Vorinostat” to “vorinostat” (line 387, in red).
  • “Bevacizumab” in “bevacizumab” (line 469, in red).

Point 3: Some punctuation marks are inappropriately inserted in the text.

Response 3: As kindly suggested, some inappropriate punctuation marks has been deleted (lines, 128, 227, 230, 266, 299, 339, 391)

Point 4: Acronyms should be always reported only the first time they are present and then used throughout text and tables. Authors are invited to check all acronyms used and to correct them.

Response 4: As kindly suggested, we have checked all acronyms in the text: we have reported the acronyms the first time and then we have used them throughout the text.  We have done the same procedure for the tables.

Point 5: Line 303…it is not specified what “Mab” is

Response 5: As kindly suggested, we have specified what Mab is (monoclonal antibodies, line 326)

Point 6: Line 51 …. the word “develepments” should be “developments”

Response 6: As kindly suggested, we have changed “develepments” to “developments” (line 51)

Point 7: Line 110 should be “…form the sympathetic nervous system”

Response 7: As kindly suggested, we have written “form the sympathetic nervous system” (line 113).

Point 8: Line 111 at the beginning of the line should be “Iodine -labeled MIBG is a functional agent…”

Response 8: As kindly suggested, we have written “Iodine -labeled MIBG is a functional agent…” (line 114).

Point 9: Line 115 the word “neoplasias” should be “neoplasia”.

Response 9: As kindly suggested, we have changed “neoplasias” to “neoplasia” (line 118)

Point 10: Line 131 the word “children” should be “child”.

Response 10: As kindly suggested, we have changed “children” to “child” (line 134)

Point 11: Line 141 the word “too” should be “to”

Response 11: As kindly suggested, we have changed “children” to “child” (line 118)

Point 12: Line 141 the word “too” should be “to”

Response 12: As kindly suggested, we have changed “too” to “to” (line 146)

Point 13: Line 142 the words “extraordinary evidence” should be “extremely rare evidence

Response 13: As kindly suggested, we have changed “extraordinary evidence” to “extremely rare evidence” (line 148).

Point 14: Line 187 the word “metastatization for” should be “metastates of”

Response 14: As kindly suggested, we have changed “metastatization for” to “metastases of” (line 204).

Point 15: Line 199 the word “neuroblastomas” should be “neuroblastoma”

Response 15: As kindly suggested, we have changed “neuroblastomas” to “neuroblastoma” (line 216).

Point 16: Line 302 the word “unlabelled” should be “unlabeled” (this misspelled word is repeated in line 302)

Response 16: As kindly suggested, we have changed “unlabelled” to “unlabeled” (line 325 and 326).

Point 17: Line 415 the word “radiolabelled” should be “radiolabeled” (this misspelled word is repeated may times)

Response 17: As kindly suggested, we have changed “radiolabelled” to “radiolabeled” (lines 356, 360, 441, 458 and in table 1).

Point 18: Table 1: In legend the word “theranostic should be “theranostics”

Response 18: the reviewer number 1 preferred the words “theragnostic” and theragnostics” instead of “theranostic” and “theranostics”, so we have changed the word “theranostic” in “theragnostics”(legend of table 1).

Point 19: Table 1: In “reference 64” the word “mousse” should be “mice”

Response 19: As kindly suggested, we have changed “mousse” to “mice” (study [67]).

Point 20: Figure 1: In legend the word “children” should be “child”

Response 20: As kindly suggested, we have changed “children” to “child”